# Evaluation of *GeneXpert vanA/vanB* in the early diagnosis of vancomycin-resistant enterococci infection

Zhuo-Lei Li[1,2], Qi-Bing Luo[1,2], Shan-Shan Xiao[1,2], Ze-Hong Lin[1,3], Ye-Ling Liu[1,4], Meng-Yi Han[1,4], Jing-Hua Zhong[1,4], Tian-Xing Ji[5], Xu-Guang Guo[1,4,6,7]*

1 Department of Clinical Laboratory Medicine, The Third Affiliated Hospital of Guangzhou Medical University, Guangzhou, China, 2 Department of Clinical Medicine, The Second Clinical School of Guangzhou Medical University, Guangzhou, China, 3 Department of Pharmacy, Guangzhou Medical University School of Pharmaceutical Sciences, Guangzhou, China, 4 Department of Clinical Medicine, The Third Clinical School of Guangzhou Medical University, Guangzhou, China, 5 Department of Clinical Laboratory Medicine, The Second Affiliated Hospital of Guangzhou Medical University, Guangzhou, China, 6 Department of Key Laboratory for Major Obstetric Diseases of Guangdong Province, The Third Clinical School of Guangzhou Medical University, Guangzhou, China, 7 Department of Key Laboratory of Reproduction and Genetics of Guangdong Higher Education Institutes, The Third Clinical School of Guangzhou Medical University, Guangzhou, China

* gysygxg@gmail.com

## Abstract

### Purpose

*Vancomycin-resistant enterococci* infection is a worrying worldwide clinical problem. To evaluate the accuracy of *GeneXpert vanA/vanB* in the diagnosis of VRE, we conducted a systematic review in the study.

### Methods

Experimental data were extracted from publications until May 03 2021 related to the diagnostic accuracy of *GeneXpert vanA/vanB* for *VRE* in PubMed, Embase, Web of Science and the Cochrane Library. The accuracy of *GeneXpert vanA/vanB* for *VRE* was evaluated using summary receiver to operate characteristic curve, pooled sensitivity, pooled specificity, positive likelihood ratio, negative likelihood ratio, and diagnostic odds ratio.

### Results

8 publications were divided into 3 groups according to two golden standard references, *vanA* and *vanB* group, *vanA* group, *vanB* group, including 6 researches, 5 researches and 5 researches, respectively. The pooled sensitivity and specificity of group *vanA* and *vanB* were 0.96 (95% CI, 0.93–0.98) and 0.90 (95% CI, 0.88–0.91) respectively. The DOR was 440.77 (95% CI, 37.92–5123.55). The pooled sensitivity and specificity of group *vanA* were 0.86 (95% CI, 0.81–0.90) and 0.99 (95% CI, 0.99–0.99) respectively, and those of group *vanB* were 0.85 (95% CI, 0.63–0.97) and 0.82 (95% CI, 0.80–0.83) respectively.

**Data Availability Statement:** The datasets generated during and/or analysed during the current study are available in the PubMed (https://pubmed.ncbi.nlm.nih.gov/), Embase (https://www.

embase.com/), Web of Science (https://www.webofscience.com/) and the Cochrane Library (https://www.cochranelibrary.com/).

**Funding:** The author(s) received no specific funding for this work.

**Competing interests:** The authors have declared that no competing interests exist.

## Conclusion

*GeneXpert vanA/vanB* can diagnose *VRE* with high-accuracy and shows greater accuracy in diagnosing *vanA*.

### Author summary

*Vancomycin-resistant enterococci* (VRE), firstly identified in the mid-1980, is a type of antimicrobial resistance bacteria. In recent years, they were found more colonization in patients with critical diseases, showing new resistance to many antibacterial drugs, which is a worrisome clinical problem worldwide. Traditionally, VRE testing is performed mainly by culture which is a standard reference but requires complex steps and takes a long time. Currently, *GeneXpert vanA/vanB* were approved as a rapid and sensitive molecular assay for detecting VRE. However, the accuracy of *GeneXpert vanA/vanB* is without systematic-analyses in evidence-based medicine. Therefore, we conducted a data integration and analysis in this study. Finally, we draw a conclusion that *GeneXpert vanA/vanB* has a high accuracy diagnosing VRE in comparison with conventional culture and PCR. Furthermore, *GeneXpert vanA/vanB* shows more accuracy in diagnosing *vanA*. In addition, we suggest that an additional test is needed for further detecting *vanB*. This finding provides a promising direction for the diagnosis of VRE to a certain extent.

## Introduction

Since 1988, *vancomycin-resistant enterococci* (*VRE*) have been found in patients with critical diseases due to extensive use of antibiotics, prolonged hospital stays and intensive care unit (ICU) admission [1]. They became a type of antimicrobial resistance (*AMR*) bacteria that most commonly spread in medical institutions, especially in Europe [2], with an incidence of 2–34.9% [3]. At present, *VRE* is prevalent globally, and its prevalence has increased significantly, which is a worrisome clinical problem worldwide [4].

*VRE* testing is currently performed mainly by traditional culture and Polymerase Chain Reaction (PCR) detection of the resistance genes *vanA* and *vanB* [5,6]. Although culture is the confirmed reference method [7,8], it takes a long time, requires complex extraction and detection steps and has a high economic impact during a *VRE* outbreak [9]. The U.S. Food and Drug Administration (FDA) approved a rapid molecular assay, the *GeneXpert vanA/vanB* [8,10], which is a unique and completely automated process that includes deoxyribonucleic acid (DNA) extraction, amplification and detection using real-time PCR. Furthermore, results are usually available in less than one hour [4,5].

It is indicated that *GeneXpert vanA/vanB* testing is sensitive as well as cost-effective [5,11]. In addition, there are some researches supporting that some indetermination results exist in that of *GeneXpert vanA/vanB* detecting van B [10,12]. There are few systematic-analyses on the diagnostic accuracy of *GeneXpert vanA/vanB* for *VRE* in evidence-based medicine. Therefore, to appraise the accuracy of *GeneXpert vanA/vanB* in the diagnosis of VRE and distinguish the differences between *GeneXpert vanA/vanB* detecting vanA and vanB, we conducted data integration and analysis.

## Material and methods

### Search strategy

A systematic literature search was carried out for publications until May 03, 2021, related to the diagnostic accuracy of *GeneXpert vanA/vanB* for *VRE*. Four databases were involved: PubMed, Embase, Web of Science and the Cochrane Library. According to PCIO criteria, the search stratagem utilized was as follows: (((*Enterococcus*) AND (*Vancomycin Resistance*)) OR (*Vancomycin-Resistant Enterococci*)) AND (*GeneXpert vanA/vanB*). Possible matches were also retrieved from the related references and the language was restricted to English.

### Study selection

Inclusion criteria:

(i) Each included study used *GeneXpert VanA/VanB* for detection of *VRE*. Clinical specimens were identified as *VRE* or standard strains by reference methods, which were regarded as the gold standards;

(ii) Human samples were detected and analyzed;

(iii) A $2 \times 2$ table was constructed with sufficient data to estimate sensitivity, specificity, and the likelihood ratio.

Exclusion criteria:

(i) Samples from animals or other species;

(ii) Reference standards cannot be found;

(iii) Incomplete raw data: when the raw data were unable to construct the $2 \times 2$ tables, or when raw data were unable to obtained from the authors;

(iv) Duplicate publications;

(v) Reviews, conference abstracts, case reports and studies that data extraction was impossible to perform.

Two independent reviewers assessed the studies according to the defined criteria above. If the results were found to be inconsistent, the third investigator was consulted and concluded the same.

### Data extraction

An Excel spreadsheet was created to collect data, which was extracted by two investigators who scanned the included literature independently. Any disagreements were reconciled by a third team member. The following variables comprise the first author's name, the publication year, the area where the research was implemented, type of study, clinical features and settings, the specimen type, reference standard test, and false and true positives and negatives (TP, TN, FP, FN). When we discuss *vanA* and *vanB* simultaneously, named *vanA* and *vanB* group, Mycobacterial culture was defined as the gold standard. When we discuss *vanA* or *vanB* separately, named *vanA* group and *vanB* group, the golden standard was defined as mycobacterial culture and PCR.

In the studied texts, multiple groups and different backgrounds were considered discrete units of analysis comprising a single study.

## Quality assessment

The quality of the publications were assessed using Quality Assessment of Diagnostic Accuracy Studies (QUADAS-2) [12]. There are four key domains that compose the tool, patient selection, the index test, reference standard and flow and timing, that evaluates bias and utility of the reviewed studies. Values of high, unclear, or low risk were assigned to grade each group of data conducted by different researchers independently figuring out the questions of the four domains. When a divergence appeared, a third investigator was invited to make the final decision.

## Statistical analysis

**(1) Statistical testing.** The pooled sensitivity, specificity, positive likelihood ratios (PLR), negative likelihood ratios (NLR), diagnostic odds ratio (DOR) and 95% confidence intervals (95% CI) were analyzed based on the data provided in the article and evaluated by forest plots, adopting a random- effects model. A value of 0.5 was added to studies with zero values to correct for continuity. A Fagan's nomogram was facilitated to estimate the clinical application of *GeneXpert vanA/ vanB* for the clinical diagnosing of *VRE* [13] by calculating the pre-test and post-test probabilities.

**(2) Analysis of heterogeneity.** In diagnostic experiments, the threshold effect or nonthreshold effect might be the primary cause of heterogeneity [14]. We gave priority to ensure whether the threshold effect exists by plotting summary receiver operator characteristic (SROC) curve and further calculating the Spearman correlation coefficient (R). An SROC space shows a typical "shoulder arm" pattern, suggesting the presence of a threshold effect. An $R \geq 0.6$ revealed a threshold effect, which manifests a rapid increase of the logit of sensitivity with the logit of 1-specificity adding [15].

Several reasons other than threshold have contributed to the appearance of correlation between sensitivity and specificity [16]. Cochran's Q test and the inconsistence index (I2) were facilitated to evaluate heterogeneity. When I2<50%, evidence shows no significant heterogeneity, use fixed- effects model. On the contrary, the random- effects model is adopted [17]. We performed meta regression and the sensitivity analysis to investigate potential sources of heterogeneity. AUC (the area under the SROC curve) takes values between 0 and 1, presenting an overall summary performance of studies [18]. To analyze publication bias, Deeks' funnel plot was applied; $P > 0.05$ showed that this meta-analysis has no publication bias [19].

**(3) Tools.** Meta-DiSc 1.4 was employed to analyze all data and STATA 12.0 was employed to draw Fagan's nomogram, bivariate box plot, and evaluating publication bias. Review Manager (RevMan) 5.3 software was applied to conduct the quality assessment.

# Result

## Publications retrieved

There are 53 published studies initially gleaned from the databases Embase (20), Web of Science (18), PubMed (15) and the Cochrane Library (0), of which 24 were left after removing duplicates. According to the titles and abstracts, 8 articles were eliminated. 8 articles were further excluded according to the exclusion criteria, through the full-text review (S1 Fig). Shows the additional reasons for exclusion. Finally, 8 publications [3,5,7,10,11,20,21] satisfied the inclusion criteria. We grouped the involving studies according to two golden standard references, named vanA and vanB group [7,8,10,11,21], vanA group [3,5,7,20,21], vanB group [3,5,7,20,21].

## Description of meta-analyzed publications

Of the 8 articles, the publication years range from 2010 to 2019. Two were from the U.S. Three studies were retrospective while the remaining were prospective. The sample size was

comprised 3064 subjects in total, 1563 subjects of which were categorized as vanA and vanB group and 2362 subjects were categorized as vanA group, vanB group. Sample types included rectal swabs, blood cultures, perianal swabs, and stool. Except for the articles which did not refer to the patients, three studies introduced patients from ICUs, one study's patients suffered from renal dialysis and another's patients were from hematology or gastroenterology departments. All bacteria were diagnosed as *VRE*.

Study characteristics in Table 1 show individual studies and their characteristics respectively.

## Heterogeneity and publication bias

No "shoulder arm" SROC curve was observed (Fig 1), and the Spearman correlation coefficient (R) was −0.943. In conclusion, there was no evidence of threshold effect. A forest map of DOR (Fig 2A) revealed that Cochran's Q = 32.40, P $\leq$ 0.01 and I$^2$ = 84.6%, indicating that significant heterogeneity was observed in the included studies. The result of meta regression (Table 2) indicated that sample types might be one possible source of heterogeneity. Sensitivity analysis showed that removal of any study did not alter the significance of the pooled effect size except the study of Zabicka (S2 Fig). After excluding this study, the I$^2$ value for heterogeneity decreased to 45% (Fig 2B). According to the method described above, Deeks' funnel plot showed no substantial asymmetry (P = 0.279). Therefore, publication bias was excluded (Fig 3).

## Methodological quality

Using RevMan 5.3, the overall methodological quality of the included studies is shown in Fig 4. Patient selection and the index test mainly contribute to the risk of bias. In patient selection domain, we assessed four studies as taking a high risk for bias, because they didn't enroll participants randomly or consecutively, and one had a case-control design [9]. In the field of the index test, two studies were assessed to be high risk for bias: one index test did not use a prespecified threshold, and the other was explained with prior knowledge of the reference standard results. In the reference standard area, most studies had a low risk of bias, as they stated

**Table 1. Basic characteristics of included studies [3,5,7,8,10,11,20,21].**

| Author | Year | Study design | Country | Sample size (No. of patients) | Clinical features and settings | Reference Standard | Specimen type |
|---|---|---|---|---|---|---|---|
| Both [10] | 2019 | Retrospective | Germany | 33(-) | _b | culture | blood cultures |
| Both [10] | 2019 | Prospective | Germany | 205(-) | _b | culture | blood cultures |
| Marner [11] | 2011 | Retrospective | America | 184(145) | Patients | culture | perianal swabs |
| Babady [8] | 2012 | Prospective | America | 300(162) | patients in bone marrow transplant units | culture | rectal swabs |
| Zabicka [21] | 2011 | Prospective | Poland | 37(37) | Patients from Hematology or gastroenterology | 1.culture 2. PCR | stool samples |
| Bourdon [7] | 2010 | Prospective | France | 804(794) | Patients | 1.culture 2. PCR | rectal swabs |
| Goossens [20] | 2011 | Prospective | Belgium | 50(-) | patients undergoing renal dialysis | 1.Culture 2.PCR | stool samples |
| Holzknecht [5] | 2017 | Prospective | Denmark | 1099(804) | patients | 1.culture 2. PCR | rectal swabs |
| Olivgeeris [3] | 2016 | Retrospective | Greece | 372(-) | patients in ICU | 1.culture 2.PCR | rectal swabs |

a: MIC: minimum inhibitory concentration.

b: No mention of clinical features.

c: No mention of MIC.

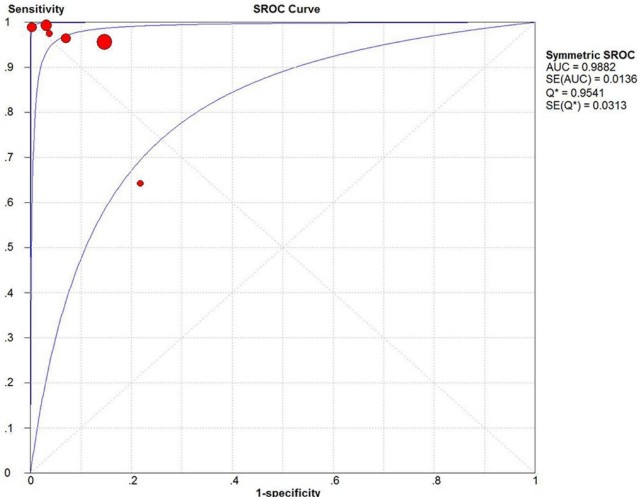

**Fig 1. Summary receiver operating curves of vanA and vanB group.** The SROC AUC was 0.9882, which is close to 1, indicating a high ability for *VRE* detection.

that the results of the reference standard were interpreted without knowing the index test results. Judging from the index test, the flow and timing of the risk of bias were relatively low. There was no concern about the assessment of applicability for nine studies in the patient selection, the index test and reference standard domain.

## Merge analysis results

The pooled sensitivity, specificity, PLR, NLR and DOR of *GeneXpert VanA/VanB* of each group were shown in Table 3. The pooled sensitivity and specificity were 0.96 (95% CI, 0.93–0.98), 0.90 (95% CI, 0.88–0.91) for vanA and vanB group, 0.86 (95% CI, 0.81–0.90) and 0.99 (95% CI, 0.99–0.99) for vanA group, 0.85 (95% CI, 0.63–0.97) and 0.82 (95% CI, 0.80–0.83) for vanB group, respectively (Fig 5).

As Fagan's nomogram showed, when the pre-test probability was set to 50%, the PLR of the upper diagonal was 24 and the post-test probability was 96%. Correspondingly, the NLR of the lower diagonal was 0.01 and the post-test probability was 1% (Fig 6).

## Discussion

To the best of our knowledge, this is the first meta-analysis accessing the overall diagnostic accuracy of *GeneXpert vanA/vanB*. In this study, we did a thorough search using strict

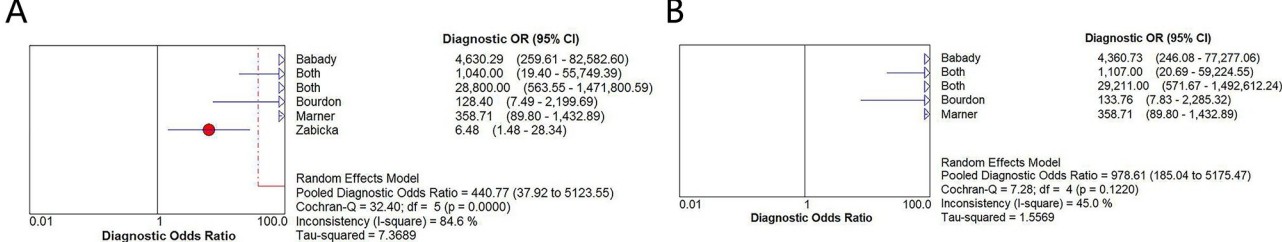

**Fig 2. Forest plot for the pooled diagnostic odds ratio of vanA and vanB group.** A Forest plot for DOR among 6 studies. B Forest plot for DOR among 5 studies (outlier study was excluded). After excluding the study, the I² value for heterogeneity decreased from 84.6% to 45%.

**Table 2. Possible sources of heterogeneity in the meta-regression analysis.**

|  | Coef | p | 95%CI |
|---|---|---|---|
| Specimen type | -1.499 | 0.023 | (-2.615, -0.384) |
| Study design | 0.418 | 0.609 | (-1.919, 2.756) |

Coef: Coefficent

screening criteria, and finally, including 8 articles, groups in different reference standards. The results of our study indicate that *GeneXpert vanA/vanB* assay has a high diagnostic accuracy. Its excellent sensitivity (0.96, 95% CI, 0.93–0.98), specificity (0.90, 95% CI, 0.88–0.91) and DOR (440.77, 95% CI, 37.92–5123.55) made it an attractive option for routine surveillance of *VRE* in the future. The combined PLR and NLR were 16.44 (95%CI, 3.66–73.86) and 0.04 (95%CI, 0.00–0.32), respectively, suggesting that *GeneXpert vanA/vanB* has a brilliant capacity to diagnose and exclude a *VRE*. The SROC AUC was 0.9882, which is close to 1, indicating a high ability for *VRE* detection. Thus, *GeneXpert vanA/vanB* showed a very good diagnostic accuracy. Fagan's nomogram showed the clinical application value of *GeneXpert vanA/vanB* in various situations.

We also conduct a study on *GeneXpert vanA/vanB* diagnosis discrepancy between *vanA* and *vanB*. The combined sensitivity, specificity, PLR, NLR, DOR of the vanA group were higher than those of the vanB group. Furthermore, the pooled NLR was lower, revealing *GeneXpert vanA/vanB* is more accurate in diagnosis on *vanA*.

That there were more false-positive results in vanB group may be attributed to the presence of genes in several species of aerobic and anaerobic bacteria that were highly similar to the *vanB* sequences [5,7]. It is inevitable for the reason these bacteria also exist in the human [21]. The culture method for all clinical E. faecium isolates may neither be feasible nor cost-efficient in the setting of every routine lab, which makes it impossible to make a clear decision about the need to isolate the patient. Hence, supplementary tests are needed for further investigating [22,23].

Sensitivity analysis demonstrated that the study of Zabicka contributes to heterogeneity. It could be influenced by the factor that the experiment performed during a VanA E. faecium outbreak, as the report of Dekeyser et al. [24], and none of the patients was colonized with VanB enterococci. Several FP vanB results may be concerned with the specimen type, stool

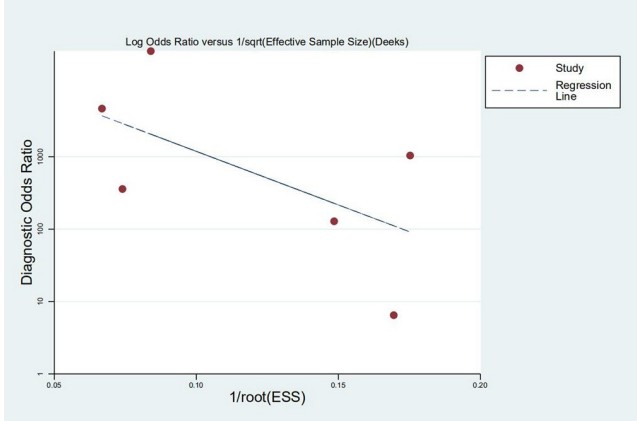

**Fig 3. Deeks' funnel plot asymmetry test of vanA and vanB group.** P = 0.279 means no Publication bias.

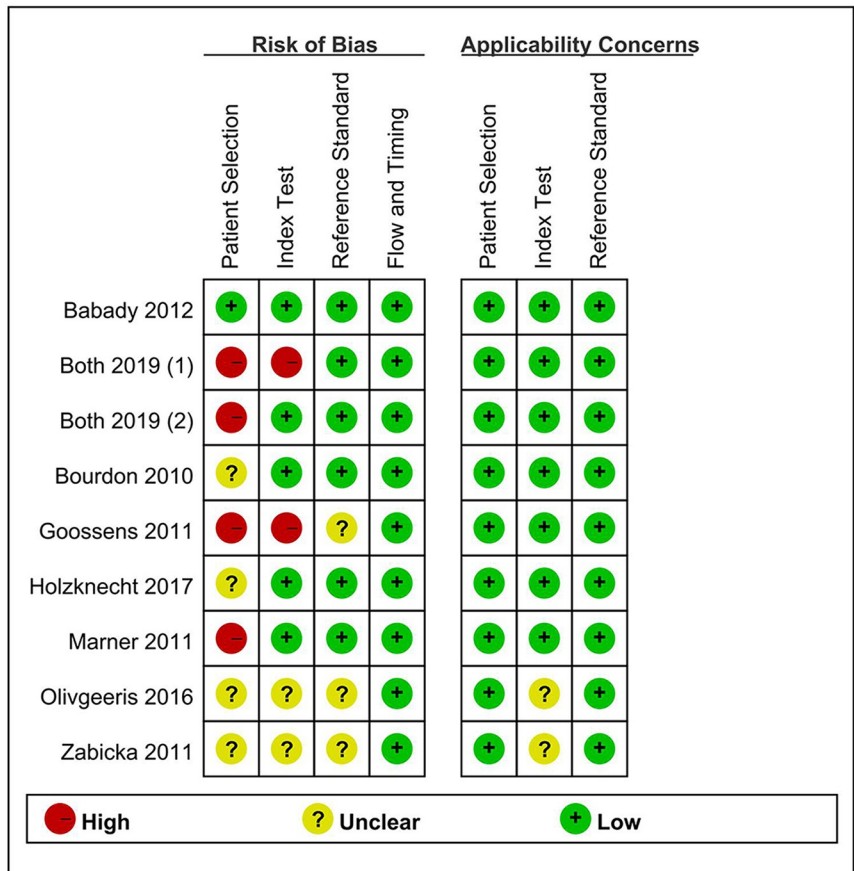

**Fig 4. Quality assessment using QUADAS-2 tool for included studies.**

**Table 3. Summarized results of the analysis.**

| Group | vanA and vanB | | | | vanA | | | | VanB | | | |
|---|---|---|---|---|---|---|---|---|---|---|---|---|
| | TP* | FP* | FN* | TN* | TP | FP | FN | TN | TP | FP | FN | TN |
| Babady [8] | 74 | 7 | 0 | 219 | - | - | - | - | - | - | - | - |
| Both [10] | 20 | 0 | 0 | 13 | - | - | - | - | - | - | - | - |
| Both [10] | 45 | 0 | 0 | 160 | - | - | - | - | - | - | - | - |
| Marner [11] | 81 | 7 | 3 | 93 | - | - | - | - | - | - | - | - |
| Bourdon [7] | 11 | 116 | 0 | 677 | 8 | 4 | 0 | 792 | 3 | 112 | 0 | 689 |
| Zabicka [21] | 9 | 5 | 5 | 18 | 8 | 3 | 5 | 21 | 0 | 6 | 0 | 31 |
| Goossens [20] | - | - | - | - | 14 | 5 | 6 | 25 | 2 | 41 | 1 | 6 |
| Holzknecht [5] | - | - | - | - | 145 | 7 | 22 | 925 | 1 | 246 | 0 | 852 |
| Olivgeeris [3] | - | - | - | - | 39 | 1 | 0 | 332 | 11 | 26 | 0 | 335 |
| Pool sensitivity(95%CI) | 0.96(0.93–0.98) | | | | 0.86(0.81–0.90) | | | | 0.85(0.63–0.97) | | | |
| Pool specificity(95%CI) | 0.90(0.88–0.91) | | | | 0.99(0.99–0.99) | | | | 0.82(0.80–0.83) | | | |
| PLR (95%CI) | 16.44(3.66–73.86) | | | | 40.61(6.74–244.53) | | | | 3.73(1.15–12.09) | | | |
| NLR (95%CI) | 0.04(0.00–0.32) | | | | 0.18(0.07–0.47) | | | | 0.40(0.08–2.16) | | | |
| DOR (95%CI) | 440.77(37.92–5123.55) | | | | 301.18(20.72–4377.94) | | | | 10.05(0.77–131.68) | | | |

Legend: -: Data was not provided in articles; TP, true positive; FP, false positive; FN, false negative; TN, true negative; PLR, positive likelihood ratios; NLR, negative likelihood ratios; DOR, diagnostic odds ratio.

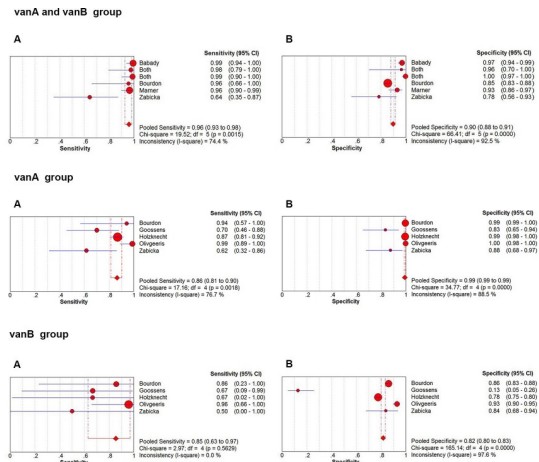

**Fig 5. Forest plots for the pooled sensitivity and specificity of three groups.** A: sensitivity B: specificity.

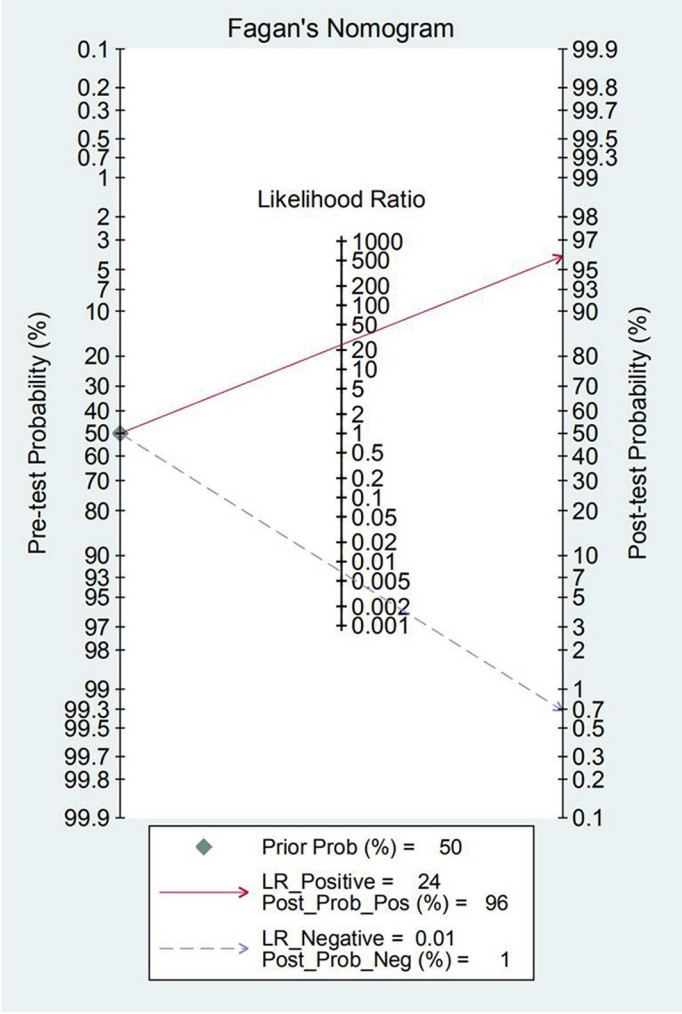

**Fig 6. Fagan's nomogram plot analysis for evaluating clinical application value.**

swabs. Stool and rectal swabs might be the harbors where anaerobic microbes were commonly checked, which increased the risks of detecting false-positive vanB results [11]. The meta regression also confirmed the specimen types might be one of possible sources of heterogeneity. The discrepancies between *GeneXpert vanA/vanB* detecting vanA and vanB might be a source of heterogeneity. Restrained by only two studies conducting both experiments on vanA and vanB detecting simultaneously or separately, a further analysis is required for more data.

There were still other variables that required to be explored, such as relevant description of patients. The sources and characteristics of patients were quite distinguished. However, sources of heterogeneity could not be formally explored for most tests because few studies were available for further evaluation.

The present study has several limitations. First, remarkable heterogeneity was observed in this study. To account for this heterogeneity, a random effects model was used to synthesis the identified studies in our analysis, which potentially increases the probability of type I error. Moreover, the results of meta regression and the sensitivity analysis were attempted to explain that detected sample could partly explain the source of heterogeneity. Subgroup analysis is looking forward to with more updating data. Second, our study also confirmed the observation of other authors that the *GeneXpert vanA/vanB* test has a low positive predictive value (PPV) for vanB enterococci [25]. Combining additional detection technologies may represent a pragmatic solution to increase VRE detection rates. Finally, we only retrieved published literature from four English databases. Only included studies written in English may have affected our findings. Despite comprehensive searches, the influence of unpublished positive results on the overall results could not be eliminated.

## Conclusion

In summary, *GeneXpert vanA/vanB* has a high accuracy diagnosing *VRE*. Furthermore, *GeneXpert vanA/vanB* shows more accuracy when diagnosing vanA. Additional test is needed for further detecting VanB.

## Supporting information

**S1 Fig. Flow chart for article search.**
(PDF)

**S2 Fig. Sensitivity analysis of each study.** Sensitivity analyses showed that removal of any study did not alter the significance of the pooled effect size except the study of Zabicka.
(TIF)

**S1 Text. PRISMA checklist.**
(DOC)

## Acknowledgments

We acknowledge PubMed, Embase, Web of Science and the Cochrane Library for providing their platforms and contributors for uploading their meaningful datasets.

## Author Contributions

**Conceptualization:** Zhuo-Lei Li, Ye-Ling Liu, Xu-Guang Guo.

**Formal analysis:** Zhuo-Lei Li, Qi-Bing Luo, Shan-Shan Xiao, Ze-Hong Lin.

**Investigation:** Zhuo-Lei Li, Qi-Bing Luo, Shan-Shan Xiao, Ze-Hong Lin.

**Methodology:** Zhuo-Lei Li, Ye-Ling Liu, Xu-Guang Guo.

**Supervision:** Tian-Xing Ji, Xu-Guang Guo.

**Validation:** Zhuo-Lei Li, Tian-Xing Ji.

**Visualization:** Zhuo-Lei Li, Qi-Bing Luo, Shan-Shan Xiao.

**Writing – original draft:** Zhuo-Lei Li, Qi-Bing Luo, Shan-Shan Xiao.

**Writing – review & editing:** Ze-Hong Lin, Meng-Yi Han, Jing-Hua Zhong.

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
