## [Decision Letter · Decision Letter 0]

26 Apr 2021

Dear Mr. Guo,

Thank you very much for submitting your manuscript "Evaluation of GeneXpert vanA/vanB in the early diagnosis of vancomycin-resistant enterococci" for consideration at PLOS Neglected Tropical Diseases. As with all papers reviewed by the journal, your manuscript was reviewed by members of the editorial board and by several independent reviewers. In light of the reviews (below this email), we would like to invite the resubmission of a significantly-revised version that takes into account the reviewers' comments. 

We cannot make any decision about publication until we have seen the revised manuscript and your response to the reviewers' comments. Your revised manuscript is also likely to be sent to reviewers for further evaluation.

Sincerely,

Alfredo G Torres

Deputy Editor

Alfredo Torres

Deputy Editor

Reviewer's Responses to Questions

**Key Review Criteria Required for Acceptance?**

**Methods**

-Are the objectives of the study clearly articulated with a clear testable hypothesis stated?

-Is the study design appropriate to address the stated objectives?

-Is the population clearly described and appropriate for the hypothesis being tested?

-Is the sample size sufficient to ensure adequate power to address the hypothesis being tested?

-Were correct statistical analysis used to support conclusions?

-Are there concerns about ethical or regulatory requirements being met?

Reviewer #1: The methods are not clearly articulated. The study is based on a literature review with no explanation as to how the raw data (if used) was acquired. Based on the bias's and the rejection criteria the sample size is at best 2 studies, not only is this is not a sufficient sample size the techniques used by the researchers of these studies vary introducing significant bias in the analysis presented in this paper. The introduction does not do a great job of setting up the problem statement and "hows" of study

Reviewer #2: study design is appropriate and the statistical analysis is accurate.

Reviewer #3: Objective were clearly stated and study design was appropriate. However the population was not clearly defined as it was given as a range from the various studies.

Not satisfied with the final number of publications used for the analysis

Appropriate Statistical analysis methods were used

**Results**

-Does the analysis presented match the analysis plan?

-Are the results clearly and completely presented?

-Are the figures (Tables, Images) of sufficient quality for clarity?

Reviewer #1: Table 1 is too large and spans multiple pages, was hard to follow

Figure Captions are insufficient

Reviewer #2: Figures can be improved.

Reviewer #3: Results were clearly and completely presented. Table 1 could be presented on a landscape page for clarity

**Conclusions**

-Are the conclusions supported by the data presented?

-Are the limitations of analysis clearly described?

-Do the authors discuss how these data can be helpful to advance our understanding of the topic under study?

-Is public health relevance addressed?

Reviewer #1: Based on the initial comments regarding data assessment, quality and bias, it is challenging to say the conclusions are supported. All the limitations are not addressed.

Reviewer #2: Authos confirmed diagnostic accurancy of GeneXpert vanA/vanB when diagnosing Vancomycin-resistant enterococci infections. in addition, Vancomycin-resistant enterococci test shows more accuracy when diagnosing van A.

As already stated by authors, the main limitations of the study are that these results are based on published data and litterature where accurancy of GeneXpert vanA/vanB has been already showed and the limited sample size. 

Auhors should better describe the implication of these results for the public health. authors suggest major caution or additional test for detecting Van B?

Reviewer #3: Limitations of the study were clearly articulated and conclusions made were appropriate

**Editorial and Data Presentation Modifications?**

Reviewer #1: (No Response)

Reviewer #2: -Abstract is not clearly presented. Authors should better specify that their study is not a de novo evaluation of the accuracy of GeneXpert vanA/vanB in the diagnosis of VRE, but rather than a data integration analysis of diffrent dataset to test GeneXpert vanA/vanB accuracy and specificty in detecting Van A and Van B.

-Extensive english editing is required throughout the entire manuscript.

Reviewer #3: Minor Revision

line 27 replace were with was

line 36 Confidence interval (0.99-0.99) not stated as a range

line 37 delete And at the start of the sentence

line 231-233-rephrase statement, too many and

**Summary and General Comments**

Reviewer #1: (No Response)

Reviewer #2: The manuscript is well structured. Results obtained from the authors could be expected investigating the accurancy of an FDA approved diagnostic test for VRE. To this purpose I suggest authors to better describe the aim of the study. Why it is important to test Van A and Van B specificity and distinctive diagnostic accuracy if test is already approved? Results obtained for Van B are enought to suggest a complementary test to be included ?

Reviewer #3: none

PLOS authors have the option to publish the peer review history of their article (what does this mean?). If published, this will include your full peer review and any attached files.

Reviewer #1: No

Reviewer #2: No

Reviewer #3: No
---

## [Decision Letter · Decision Letter 1]

10 Aug 2021

Dear Mr. Guo,

Thank you very much for submitting your manuscript "Evaluation of GeneXpert vanA/vanB in the early diagnosis of vancomycin-resistant enterococci infection" for consideration at PLOS Neglected Tropical Diseases. As with all papers reviewed by the journal, your manuscript was reviewed by members of the editorial board and by several independent reviewers. The reviewers appreciated the attention to an important topic. Based on the reviews, we are likely to accept this manuscript for publication, providing that you modify the manuscript according to the review recommendations. 

Sincerely,

Alfredo G Torres

Deputy Editor

Alfredo Torres

Deputy Editor

Reviewer's Responses to Questions

**Key Review Criteria Required for Acceptance?**

**Methods**

-Are the objectives of the study clearly articulated with a clear testable hypothesis stated?

-Is the study design appropriate to address the stated objectives?

-Is the population clearly described and appropriate for the hypothesis being tested?

-Is the sample size sufficient to ensure adequate power to address the hypothesis being tested?

-Were correct statistical analysis used to support conclusions?

-Are there concerns about ethical or regulatory requirements being met?

Reviewer #2: (No Response)

**Results**

-Does the analysis presented match the analysis plan?

-Are the results clearly and completely presented?

-Are the figures (Tables, Images) of sufficient quality for clarity?

Reviewer #2: (No Response)

**Conclusions**

-Are the conclusions supported by the data presented?

-Are the limitations of analysis clearly described?

-Do the authors discuss how these data can be helpful to advance our understanding of the topic under study?

-Is public health relevance addressed?

Reviewer #2: (No Response)

**Editorial and Data Presentation Modifications?**

Reviewer #2: minor revision

**Summary and General Comments**

Reviewer #2: Authors provided most of the corrections required. Abstract has been improved but English editing still need to be performed. Manuscript should be revised by a native English speaker.

PLOS authors have the option to publish the peer review history of their article (what does this mean?). If published, this will include your full peer review and any attached files.

Reviewer #2: No

Figure Files:

Data Requirements:

Reproducibility:

References

---

## [Decision Letter · Decision Letter 2]

1 Oct 2021

Dear Dr. Guo,

We are pleased to inform you that your manuscript 'Evaluation of GeneXpert vanA/vanB in the early diagnosis of vancomycin-resistant enterococci infection' has been provisionally accepted for publication in PLOS Neglected Tropical Diseases.

Best regards,

Travis J Bourret

Associate Editor

Alfredo Torres

Deputy Editor

Reviewer's Responses to Questions

**Key Review Criteria Required for Acceptance?**

**Methods**

-Are the objectives of the study clearly articulated with a clear testable hypothesis stated?

-Is the study design appropriate to address the stated objectives?

-Is the population clearly described and appropriate for the hypothesis being tested?

-Is the sample size sufficient to ensure adequate power to address the hypothesis being tested?

-Were correct statistical analysis used to support conclusions?

-Are there concerns about ethical or regulatory requirements being met?

Reviewer #2: (No Response)

**Results**

-Does the analysis presented match the analysis plan?

-Are the results clearly and completely presented?

-Are the figures (Tables, Images) of sufficient quality for clarity?

Reviewer #2: (No Response)

**Conclusions**

-Are the conclusions supported by the data presented?

-Are the limitations of analysis clearly described?

-Do the authors discuss how these data can be helpful to advance our understanding of the topic under study?

-Is public health relevance addressed?

Reviewer #2: (No Response)

**Editorial and Data Presentation Modifications?**

Reviewer #2: (No Response)

**Summary and General Comments**

Reviewer #2: (No Response)

PLOS authors have the option to publish the peer review history of their article (what does this mean?). If published, this will include your full peer review and any attached files.

Reviewer #2: No

---

## [Editor Report · Acceptance letter]

23 Oct 2021

Dear Mr. Guo,

We are delighted to inform you that your manuscript, "Evaluation of GeneXpert vanA/vanB in the early diagnosis of vancomycin-resistant enterococci infection," has been formally accepted for publication in PLOS Neglected Tropical Diseases.

Best regards,

Shaden Kamhawi

co-Editor-in-Chief

Paul Brindley

co-Editor-in-Chief
